# Void Content Minimization in Vacuum Infusion (VI) via Effective Degassing

**DOI:** 10.3390/polym13172876

**Published:** 2021-08-27

**Authors:** Jaime Juan, Arlindo Silva, Jose Antonio Tornero, Jose Gámez, Nuria Salán

**Affiliations:** 1Departamento de Ciència i Enginyeria de Materials (CEM), Universitat Politècnica de Catalunya (UPC), 08222 Terrassa, Spain; jaime.juan@upc.edu; 2Engineering Product Development Pillar, Singapore University of Technology and Design, Singapore 487372, Singapore; arlindo_silva@sutd.edu.sg; 3Institut d’Investigació Tèxtil i Cooperació Industrial de Terrassa (INTEXTER), Universitat Politècnica de Catalunya (UPC), 08222 Terrassa, Spain; jose.antonio.tornero@upc.edu; 4Departamento d’Enginyeria de Sistemes Industrials i Disseny (ESID), Universitat Jaume I (UJI), 12071 Castelló de la Plana, Spain; jose.gamez@esid.uji.es

**Keywords:** Fiber Reinforced Polymer (FRP), Vacuum Infusion (VI), degassing, porosity

## Abstract

This paper addresses the major concern which component porosity represents in Vacuum Infusion (VI) manufacturing due to resin gelation at pressures close to absolute vacuum. Degassing is a fundamental step to minimize or even avoid resin outgassing and enhance dissolution of voids created during preform impregnation. The efficacy of different degassing procedures based on vacuum degassing, and assisted by adding a nucleation medium, High Speed (HS) resin stirring and/or later pressurization during different time intervals have been analyzed in terms of final void content is studied. Through a rigorous and careful design of the manufacturing process, outgassing effects on final void content were isolated from the rest of porosity causes and specimens with two clearly identifiable regions in terms of porosity were manufactured to facilitate its analysis. Maximum void content was kept under 4% and porous area size was reduced by 72% with respect to conventional vacuum degassing when resin was stirred at HS; therefore, highlighting the importance of enhancing bubble formation during degassing.

## 1. Introduction

Although Vacuum Infusion (VI) is a promising alternative to Resin Transfer Molding (RTM) and even prepreg manufacturing, it presents some drawbacks in terms of quality, such as lower fiber volume fraction, $v_{f}$, and higher void volume fraction, $v_{0}$ [1,2]. Void content, also referred to as porosity, is crucial in matrix performance. Mechanical properties of Fiber Reinforced Polymers (FRP) such as compression, inter-laminar shear and flexural strengths, and fatigue behavior are seriously affected by void content [3,4,5,6]. If porosity is extended to the surface, even surface finish of FRP components may be altered, deteriorating component aesthetics [7] and later bonding [8]. A previous paper presented a new VI process to which the current paper contributes with further studies [9].

The main causes of void formation in VI and RTM are resin flow through dual-scale heterogeneous porous media and outgassing of air dissolved into the resin. Both causes can be addressed by controlling processing conditions. Several studies are focused on predicting void formation in dual-scale porous media due to resin flow [10,11,12]; however, resin outgassing is not a common subject in FRP manufacturing research, even though it is a major concern in VI since resin is prone to outgas at pressures close to absolute vacuum due to its minimum air solubility, as stated by Henry’s law.

Although final porosity depends on the whole manufacturing process and materials involved [13,14,15], resin outgassing may be reduced or even avoided by carrying out a proper resin degassing procedure before preform impregnation. Besides, resin capacity for dissolving voids formed during preform filling depends on the previous air concentration into the resin. Conventional degassing approaches applied in the FRP field consist of exposing a volume of resin to high vacuum levels for a specific amount of time. Nevertheless, the degassing efficacy of this procedure is questionable if the physics involved in the process is considered; since air is initially dispersed as molecules into the resin and molecules are removed very slowly from the solution through the resin free surface via diffusion.

Vacuum degassing can be speeded up by creating bubbles which can be removed faster. Therefore, improving attempts are usually based on enhancing heterogeneous bubble nucleation by adding a nucleation medium or sparging [16,17]. Air molecules diffuse to bubbles, reaching a saturated solution, but are no longer over-saturated. However, some micro-bubbles may remain trapped near the resin surface after degassing; However, the combination of a nucleation medium and a system of capillary separation appears useful at filtrating these micro-bubbles [16].

A straight and reliable way of characterizing resin outgassing behavior in VI manufacturing remains challenging, since it not only depends on the quantity of dissolved air into the resin, but on manufacturing conditions and materials involved too. For example, Unifilo E-glass mat exhibits especially good bubble nucleation properties, enhancing resin outgassing [17].

In polymer manufacturing, in order to improve efficacy, degassing under vacuum pressures is often combined with auxiliary systems such as mixing, rotation, or sonication, which also help bubble formation through rupturing liquid polymers by cavitation [18].

In order to bridge the gap between VI and RTM in terms of component quality through minimizing void content, this paper explores the efficacy of conventional vacuum degassing, and the benefits of additional auxiliary systems which enhance bubble formation during degassing and dissolution of the remaining micro-bubbles in the volume of resin. Degassing effectiveness has been directly assessed by means of the resin outgassing behavior during VI manufacturing of glass-epoxy specimens, through the final specimen porosity content and after isolating outgassing effects from the rest of porosity causes. After characterizing manufactured specimens, a screening experiment, based on a fractional factorial design of experiments, was conducted to analyze the effects on specimen porosity of degassing time, addition of a nucleation medium in the volume of resin, stirring the resin at High Speed (HS) while degassing and later pressurization of the resin.

## 2. VI Manufacturing: Decision-Making

The VI process proposed in this paper was designed to minimize differences between specimens in void formation due to flow through dual-scale porous media, and to promote a gradient of resin pressure into preforms when gelation occurred. In-plane filling of preforms, in combination with short post-filling times, should result in specimens containing decreasing gradients of thickness and pressure between inlet and venting channels. According to Henry’s law, this gradient of pressure should result in a gradient of void content in each specimen due to different outgassing conditions. Outgassing is enhanced as resin pressure decreases. Furthermore, from a specific pressure level, outgassing should not occur and porosity-free regions close to inlet channels should appear; while porosity accumulates near venting channels.

VI is a complex multi-step manufacturing process whose main steps are governed by pressure, temperature, and time. Processing parameters governing specimen manufacturing in the experiments are provided in Figure 1. The degassing parameters which were the focus of this study were degassing time, $t_{deg}\;$, and degassing auxiliary systems. As will be laid out further on, in the eight experiments, a total of four different vacuum degassing configurations were adopted: conventional (Figure 2a), assisted by adding a nucleation medium over the base of the resin pot (Figure 2b), assisted by HS resin stirring (Figure 2c), and assisted by both adding a nucleation medium and HS resin stirring (Figure 2d). These vacuum configurations were then combined with different $t_{deg}$ and the possibility of later resin pressurization prior to the filling stage. Degassing was carried out at $P_{deg}=-98\pm 1\;\mathrm{kPa}$. The nominal ultimate pressure achievable by the vacuum pump was 0.1 kPa (absolute pressure).

Preforms were enclosed in an ordinary assembly with peel-ply layers covering preform top and bottom surfaces (Figure 3). Inlet and venting channels were placed parallel to preform edges (z direction) to force a rectilinear flow front progression and in-plane preform impregnation (x direction). No separation was allowed between channels and preform edges to avoid unnecessary flow resistance; since it is a proven cause of pressure equalization into the preform [19].

Void formation at the flow front through dual-scale porous preforms is often addressed in the literature by the modified capillary number, $Ca^{*}$ [20,21]:(1)$$Ca^{*}=\frac{\mu \cdot u_{f}}{\gamma \cdot \cos\theta }$$
where $u_{f}$ is the macroscopic resin velocity at the flow front, μ is the resin dynamic viscosity, γ is the resin surface tension, and θ is the contact angle between resin and reinforcements. Furthermore, macroscopic resin velocity, u, is given by Darcy’s law:(2)$$u=-\frac{K}{\mu }\frac{dP}{dx}$$
where K is the preform permeability and dP/dx is the pressure gradient along the filled region of the preform. Both Equations (1) and (2) have been reduced to the 1D rectilinear flow case in the x direction.

Void formation is negligible in a specific range of $Ca^{*}$; while intra- and inter-tow voids are formed at higher and lower values of $Ca^{*}$, respectively (Figure 4). Therefore, void formation differences between manufactured specimens may arise from μ and u, which in turn may also vary with changes in K and/or dP/dx.

Resin viscosity, μ, is a function of temperature and time elapsed from the onset of the reaction of polymerization. As the reaction progresses, the degree of crosslinking increases, involving a continuous raise in μ and making it more difficult for the resin to flow through the preform. Initially, the crosslinking reaction advances slowly; but large variations in reaction times may cause substantial changes in μ. Not all degassing procedures took the same time; hence, although resin was kept at ambient temperature, $T_{amb}$, flow differences could arise due to different values of μ during preform filling. In spite of not directly monitoring μ along the VI process, times elapsed from resin mixing until the onset of the filling stage, $t_{0\to fill_{0}}$, and until the end of the filling stage, $t_{0\to fill_{end}}$, were recorded to account for the effects caused by μ variations.

Preform permeability, K, depends on the compressive pressure history exerted on the preform along all the successive manufacturing steps (debulking, filling and post-filling) [22,23,24,25] and, thus, is also closely connected to dP/dx. However, characterizing K during the different VI steps is a challenging task, since multi-layer textile preforms exhibit highly complex inelastic compressive behavior such as dependence on compaction velocity, stress relaxation and stress–strain hysteresis [26,27].

Furthermore, in VI, compaction and resin pressures are coupled due to the flexibility of one mold half, as pointed out in Terzaghi’s relation [28], in which the normal pressure applied to the fiber-matrix system, *P_atm_*, is decomposed into the sum of resin pressure, *P*, and fibre compaction stress, *σ_f_*:
(3)$$P_{atm}=P+\sigma_{f}$$

Since specimen materials and size were kept constant along the research, and governing pressure controlled along the test campaign; main variations in K and dP/dx could have appeared due to different debulking times, $t_{deb}$. Because debulking and degassing steps were carried out in parallel, different $t_{deg}$ involved variations in $t_{deb}$. However, debulking was planned to include a single loading step to the minimum attainable vacuum pressure, $P_{deb}$, for $t_{deb}>75\;\min$, which was considerably longer than time required to fiber settling occur (<30 min for preforms later introduced). Therefore, expected K and dP/dx variations between specimens would be caused by inherent preform variability more than by processing conditions.

Variations in u could be also evaluated through monitoring filling time, $t_{fill}$. The correlation between both variables becomes clear by an analytical expression, such as the one provided below, to determine $t_{fill}$ [29]:(4)$$t_{fill}=-\frac{\mu }{2}\;\frac{L^{2}}{{\left(\frac{K}{\phi }\frac{dP}{d\alpha }\;\right)}_{\alpha =1}}$$
where L is the filling length, α is the relative position such as $\alpha =x/x_{f}$, $x_{f}$ is the flow front position and ϕ is preform porosity (distinct from specimen porosity associated to void content). It must be noted how an alternative version of Equation (2) takes part in $t_{fill}$ calculation. Equation (4) provides $t_{fill}$ when the preform is fully filled, $x_{f}=L$.

On the other hand, resin outgassing depends on both the quantity of dissolved air into the resin and the resin capacity of dissolving air. The above listed degassing procedures tried to minimize air content into the resin; while air solubility in equilibrium is given by Henry’s law as:(5)$$C_{air}=H\cdot P_{air}$$
where $C_{air}$ is the solubility of air at a fixed temperature, $P_{air}$ is the partial pressure of air and H is Henry’s law solubility constant, which depends on temperature (decreasing with rising temperatures) and the resin.

In equilibrium, $P_{air}$ equals the resin pressure, $P=P_{air}$; hence, air solubility into the preform would be a function of the position, $P\left(x\right)$. Forcing the existence of a gradient of pressure into the specimen between inlet and venting channels at resin gelation would allow to capture a continuous distribution of air solubility conditions. After preform filling, the inlet channel was clamped while vent pressure was kept constant. Achieving homogeneous distributions of thickness and pressure would have required even a longer post-filling time, $t_{pfill}$, than $t_{fill}$ [30,31,32]; however, $t_{pfill}=30\;\min$ was set to only about half of $t_{fill}$, resulting in the expected gradients of thickness and pressure between inlet and venting channels. Post-filling was monitored through preform thickness measurement with two laser displacement sensors at approximately 30 mm from the inlet and venting channels.

Resin cure was carried out in a single cure cycle at $T_{cure}=80{\;}^{{}^{\circ}}C$ to assure a rapid gelation of the resin after the post-filling step and avoid pressure homogenization into the preform; although, it implied a reduction in air solubility. The heating source was a heating blanket placed under the mold. Assemblies were covered with a non-woven polyester fabric with a thickness of 20 mm to guarantee a homogeneous temperature distribution along specimen thickness.

It is worth noting that filling, post-filling and curing steps were conducted at pressures higher than $P_{deg}$ in order to increase resin capacity of dissolving bubbles formed during preform impregnation.

## 3. Materials and Methods

### 3.1. Specimens

Constituent materials used in the experimentation were a tight E-Glass 2/2 Twill-Weave Fabric (GTWF) Angeloni VV 320 T and a DGEBA epoxy system Sicomin SR 8100–SD 8822. This system exhibits low viscosity, 340–390 mPa·s, and shows a working time longer than 200 min at ambient temperature [33].

A total of 9 GTWF layers of 320 mm × 220 mm with two orientations, $\left(0/90\right)$ and $\left(+45/-45\right)$, were alternately stacked over a flat aluminum mold resulting in quasi-isotropic preforms ${\left[\left(0/90\right),\left(+45/-45\right)\right]}_{2},\left(0/90\right),{\left[\left(+45/-45\right),\left(0/90\right)\right]}_{2}$. Due to the irregular free-edge thickness, specimens were trimmed to a useful area, $S_{useful}$, of 300 mm × 200 mm.

### 3.2. Test Procedures

Specimen characterization was not limited to porosity-related attributes. Specimen quality was also addressed in terms of $v_{f}$. Besides, morphology and distribution of trapped pores, and their effects on the flexural response of the specimens were also studied in order to give a clear picture of the problem caused by trapped porosity.

A series of samples were systematically distributed to capture the intentionally caused gradients of fiber and void content into each specimen (Figure 5): two rows of samples parallel to the flow direction (dir. x) to measure constituent contents (CYZ) and pore morphology (MYZ), and three rows of samples perpendicular to the flow direction (dir. z) to capture flexural properties along specimens (FYZ).

#### 3.2.1. Visual Inspection

The translucent nature of the GTWF-epoxy specimens allowed to identify the specimen areas in which porosity was present. For each specimen, porous area fraction, $s_{0}$, was computed as the ratio of porous area, $S_{0}$, and specimen useful area, $S_{useful}$:(6)$$s_{0}=\frac{S_{0}}{S_{useful}}$$

The open-source image processing package Fiji, based on ImageJ, was used to scan the top view pictures of the specimens.

#### 3.2.2. Loss on Ignition Method

Samples coded as CYZ in Figure 5 were tested via the loss on ignition method, according to the procedure stated in the standards UNE-EN ISO 1172:1999 and UNE-EN ISO 7822:2001, in order to determine fiber volume fraction, $v_{f}$, and void volume fraction, $v_{0}$.

Initially, constituent materials volume fractions were computed from the mass fractions and the sample volume, $V_{sample}$, such that:(7)$$v_{f}=\frac{m_{f}/\rho_{f}}{V_{sample}}\;,v_{r}=\frac{m_{r}/\rho_{r}}{V_{sample}}$$
where $m_{f}$ and $m_{r}$ are the fibre and resin mass fractions, $v_{f}$ and $v_{r}$ are the fibre and resin volume fractions, and $\rho_{f}=2544{\;\mathrm{kg}/m}^{3}$ and $\rho_{r}=1095{\;\mathrm{kg}/m}^{3}$ are the fiber and resin densities [34]. Then $v_{0}$ was estimated for each sample by applying the balance of component fractions:(8)$$v_{0}=1-v_{f}-v_{r}$$

The determination of $v_{0}$ presented two limitations: 25 mm × 25 mm samples resulted in a range of volumes $\sim 1.4\;{\mathrm{cm}}^{3}$ lower than the minimum $2{\;\mathrm{cm}}^{3}$ recommended in the standards; and the texturized surface created by the peel-ply caused an over-estimation of sample volumes and, thus, $v_{0}>0\%$ were obtained even in samples with no porosity expectation.

Since translucent specimens allowed to identify which samples contained trapped pores, those n samples with expectation of no porosity were used to compute an average surface void volume, ${\bar{V}}_{0}^{surf}$, as:(9)$${\bar{V}}_{0}^{surf}=\frac{1}{n}\sum \left(v_{0}\cdot V_{sample}\right)$$

Then, updated fiber and resin volume fractions, $v_{f}^{’}$ and $v_{r}^{’}\;$respectively, were calculated considering an adjusted volume according to ${\bar{V}}_{0}^{surf}$, such that:(10)$$v_{f}^{’}=\frac{m_{f}/\rho_{f}}{V_{sample}-{\bar{V}}_{0}^{surf}}\;,\;v_{r}^{’}=\frac{m_{r}/\rho_{r}}{V_{sample}-{\bar{V}}_{0}^{surf}}$$

Finally, $v_{0}^{’}$ was estimated for each sample by applying the balance of component fractions:(11)$$v_{0}^{’}=1-v_{f}^{’}-v_{r}^{’}$$

#### 3.2.3. Light Microscopy

Samples coded as MYZ in Figure 5 were used to measure void content and characterize void morphology through digital image processing. Micrographs were taken from cross-sections underlined in the referred figure.

Micrographic samples were casted in polyester resin round micrographic specimens, grinded with diamond discs and finally polished in two steps, with a two-in-one suspension of 6 µm diamond and lubricant, and with a silica suspension. A number of fourteen to sixteen micrographs per sample were taken at 50× magnification and stitched to get a picture covering the full sample length. The open-source image processing package Fiji, based on ImageJ, was used to process the micrographs.

Total void content, $a_{0}$, intra-tow void content, $a_{0}^{intra}$, and inter-tow void content,$\;a_{0}^{inter}$, were calculated in the set of stitched micrographs as the ratio of the area occupied by all voids of each void type, $A_{0}$, $A_{0}^{intra}$, and $A_{0}^{inter}$ respectively, and the area occupied by the sample, $A_{Sample}$, as:(12)$$a_{0}=\frac{\sum A_{0}}{A_{sample}}\;,\;a_{0}^{intra}=\frac{\sum A_{0}^{intra}}{A_{sample}}\;,\;a_{0}^{inter}=\frac{\sum A_{0}^{inter}}{A_{sample}}$$

Additionally, Feret’s diameter, $\emptyset_{Feret}$ (maximum distance between any two points belonging to a pore), Feret’s angle, $\varphi_{Feret}$ (angle formed by the Feret’s diameter and the horizontal axis), and aspect ratio, AR, were also computed for all the voids belonging to each sample. AR was defined as the ratio of the two second moments of area of a pore around its principal axes, $i_{1}$ and $i_{2}$ (AR of a circle and a square is 1) such that:(13)$$AR=\frac{i_{1}}{i_{2}}\leq 1$$

#### 3.2.4. Three-Point Flexural Test

Flexural properties of samples coded as FYZ in Figure 5 were obtained through a three-point flexural test according to the procedure stated in the standard UNE-EN ISO 14125:1999.

### 3.3. Fractional Factorial Design

Fractional factorial designs are particularly useful in early stages of experimental work, when it is likely that many factors are involved and some may have little or no effect on the response variable [35].

The effectiveness of a total of eight vacuum degassing procedures were compared through the analysis of their effects on the porosity of the corresponding VI manufactured specimens. Each degassing procedure was defined by the level adopted by a set of four factors previously introduced (Table 1): nucleation medium (N), HS stirring (S), pressurization (P), and degassing time (T).

Factor levels were arranged according to the principal one-half fraction of a two-level fractional factorial design, $2_{\mathrm{IV}}^{4-1}$ (Table 2). In two-level factors, the effect of a factor (or interaction) is expressed as the difference between the averages of a response variable at the high (+) and low (−) levels of the factor (or interaction). For example, in case of factor T, its effect, $l_{T}$, was given by:(14)$$l_{T}={\bar{y}}_{T^{+}}-{\bar{y}}_{T^{-}}$$
where ${\bar{y}}_{T^{+}}$ and ${\bar{y}}_{T^{-}}$ are the average of a response variable at high level (+) and low level (−), respectively.

Only effects of main factors N, S, P, and T, and two-factor interactions were considered in the analysis, neglecting the influence of higher order interactions. Besides, in 2^4−1^ factorial designs, effects of two-factor interactions are aliased with each other ($l_{NS}=l_{PT}$, $l_{NP}=l_{ST}$, $l_{NT}=l_{SP}$); therefore, interaction effects were carefully addressed to discern, according to main factor effects and experimental conditions, which was the predominant interaction.

Specimen porosity was characterized by different procedures, but only attributes which showed evident variation between specimens were taken as response variables of the factorial design. This variation was assessed by the coefficient of variation (or relative standard deviation), defined as the ratio between the standard deviation and the mean response.

Since the initial fractional factorial design was saturated (no available degree of freedom to compute error variance), effect significance was firstly qualitatively assessed through a half-normal probability plot of the effects. The largest effects which did not lie along the normal straight line were considered good candidates to include in the later Analysis of Variance (ANOVA), which statistically evaluated significance of factor or interaction effects on response variables. Backward elimination was performed to sequentially remove any factor or interaction from models with a significance level p>0.10 (≡10%); although, actually, effect significance was set at p≤0.05 (≡5%).

From the rigorous analysis of the VI process previously presented, apart from the design factors considered in the fractional factorial design (N, S, P, and T) and the held-constant factors (Figure 1), a set of nuisance factors (RH, $T_{deg}$, $T_{fill}$, and $T_{pfill}$) and allow-to-vary factors ($P_{deb}$, $t_{deb}$, $P_{deg}$, $t_{0\to fill_{0}}$, $t_{0\to fill_{end}}$, and $t_{fill}$.) that might affect the response variables were identified. These factors may be treated as covariates and analyzed through a variation of ANOVA, an Analysis of Covariance (ANCOVA) [36]. Special attention was kept on filling-related times, which showed large variations depending on the degassing procedure and were directly connected to void formation due to flow through dual-scale porous media.

Statistical analyses were performed with Minitab 17 statistics software package.

## 4. Results

### 4.1. Porous Area Fraction

Photographic evidence of the eight manufactured specimens is shown in Figure 6. As planned, a porous area was formed near venting channels. On each specimen in Figure 6, porous area is highlighted by underlying its boundary and the resulting porous area fraction, $s_{0}$, is provided for each set of manufacturing conditions. It can be observed that in specimens 4 and 6, $s_{0}$ was considerably larger than in the rest of specimens.

Figure 6 also provides evidence of the existence of a decreasing gradient of thickness between the inlet and venting channels, that should lead to an increasing gradient of fiber volume fraction.

### 4.2. Fiber and Void Volume Fractions

In Figure 7, it is depicted the results of the loss on ignition method for the updated fiber volume fraction, $v_{f}^{’}$, and void volume fraction, $v_{0}^{’}$. The average surface void volume, ${\bar{V}}_{0}^{surf}$, computed to correct the thickness overestimation caused by texturized surfaces was $66\;{\mathrm{mm}}^{3}$, which is equivalent to a reduction of 0.11 mm in effective sample thickness.

Only data of samples CY1, CY4, and CY8 is shown in Figure 7a for clarity reasons, each sample representing conditions near the inlet and venting channels, and in an intermediate position. It can be observed that the average per sample of $v_{f}^{’}$ and into each specimen increased at approaching the venting channels, involving an equivalent decreasing gradient of thickness along the specimens.

In the case of non-texturized samples (flat top and bottom specimen surfaces), the fiber volume fraction could have been computed through laminate thickness, h, as:(15)$$v_{f}^{h}=\frac{n_{l}\cdot \rho_{A}}{\rho_{f}\cdot h}$$
where $n_{l}=9$ is the number of layers, $\rho_{A}=326\;g/m^{2}$ is the areal density of GTWF and $\rho_{f}=2544\;\mathrm{kg}/m^{3}$ is the E-glass fibre density [34]. However, the over-estimation of h due to the surface texture involved the underestimation of the results computed by Equation (15), as can be seen in Figure 7a. Nevertheless, h could still be used as a good estimator of $v_{f}^{’}$ through a linear model adjusted with the experimental data, such that:(16)$${\hat{v}}_{f}^{’}\left[\%\right]=14.9+\frac{84.6}{h\left[\mathrm{mm}\right]},\;R^{2}=73.8\%$$

Another gradient into each specimen of $v_{0}^{’}$ is also observable in Figure 7b. All samples that are not included in Figure 7b contain no porosity as can be checked in Figure 6. As expected, the maximum void content into each specimen occurred in the vent side, samples CY8. Besides, a considerable $v_{0}^{’}>2\%$ was measured even in the first sample belonging to the porous area into each specimen, reflecting a sudden accumulation of voids instead of a gradual increment of porosity from free-void samples.

### 4.3. Voids Size, Shape, and Spatial Distribution

Microscopy analysis was focused on samples MY8, which contained the highest void content into each specimen according to the results presented in the previous section. Since inter-tow voids were considerably larger than intra-tow voids, in order to automate their identification, a void area $A_{0}=3000\;{\mu m}^{2}$ was set as the boundary between both void types.

Although intra-tow void occurrence was higher than inter-tow’s in general; the huge difference in $A_{0}$ between both void types (Figure 8a) involved that most of the void area fraction, $a_{0}$, belonged to inter-tow voids, as shown in Figure 8b. Besides, it can be noticed that $a_{0}$ and void content measured through the loss on ignition method, $v_{0}^{’}$, differed significantly.

The heterogeneous distribution of porosity into the laminates can be seen in the micrographic samples of specimen number 4 shown in Figure 9. Inter-tow voids were predominantly formed between fabric layers. Apparently, void size depended on the local nesting between layers in each analyzed cross-section; hence, $a_{0}$ sensitivity with respect to small variations in cross-section location should be high. Microscopy analysis also allowed the observation of the fast transition between non-porous and porous areas. As can be seen in sample M45, the first sample belonging to the porous area in specimen 4 (Figure 9b), the occurrence of a few inter-tow voids directly caused the accumulation of a considerable void content.

Once analyzing inter-tow voids morphology in more depth, a significant correlation between Feret’s diameter, $\emptyset_{Feret}$, and $A_{0}^{inter}$ arose aa representing the log transformation of both features, as seen in Figure 10a. A similar trend can be noticed in Figure 10b between $\emptyset_{Feret}$ and the aspect ratio, AR; although in this case, at increasing $\emptyset_{Feret}$, AR decreased. Obviously, inter-tow void tows were oriented according to the free space between GTWF layers; therefore, at increasing $\emptyset_{Feret}$, inter-tow void orientation tended to $0^{{}^{\circ}}$ (dir. x), as shown in Figure 10c.

Pore size attributes, orientation and AR reflected the direct dependence of inter-tow void morphology from size and shape of resin rich areas between fabric layers in which inter-tow voids grew. Formation of resin rich areas depends in turn on preform properties, such as fabric architecture, relative orientation between consecutive fabric layers and nesting.

### 4.4. Flexural Response

In Figure 11, it is depicted both flexural strength, $\sigma_{fM}$, and modulus, $E_{f}$, with respect to ${\hat{v}}_{f}^{’}$, estimated through Equation (16), and $v_{0}^{’}$, at the corresponding sample CYZ (flexural sample FYZ and sample CYZ belonged to the same row of samples shown in Figure 5).

Although both $\sigma_{fM}$ and $E_{f}$ should depend on ${\hat{v}}_{f}^{’}$, due to the reduced variation of ${\hat{v}}_{f}^{’}$ along samples, it was only shown a slight increment on both flexural properties as ${\hat{v}}_{f}^{’}$ increased. On the other hand, a significant dependence of $\sigma_{fM}$ on $v_{0}^{’}$ arose as shown in Figure 11; while no relation seemed to exist between $E_{f}$ and $v_{0}^{’}$.

Two linear models were fitted with the experimental data to estimate $\sigma_{fM}$ and $E_{f}$ from material quality attributes, ${\hat{v}}_{f}^{’}$ and $v_{0}^{’}$:(17)$${\hat{\sigma }}_{fM}\left[\mathrm{MPa}\right]=-83+11.8{\hat{v}}_{f}^{’}\left[\%\right]-13.4v_{0}^{’}\left[\%\right],\;R^{2}=48.2\%$$
(18)$${\hat{E}}_{f}\left[\mathrm{GPa}\right]=-5.55+0.381{\hat{v}}_{f}^{’}\left[\%\right]-0.0551v_{0}^{’}\left[\%\right],\;R^{2}=14.5\%$$

Nevertheless, experimental variability explained by both models did not reach even 50%. In the case of $E_{f}$, a very low coefficient of determination, $R^{2}$, was obtained.

Reduction in $\sigma_{fM}$ was slightly masked by some abnormally high values. Due to the heterogeneous porosity distribution, it is coherent to expect some samples which do not show any detrimental effect on flexural performance. Although it was identified a correlation between $\sigma_{fM}$ and $v_{0}^{’}$, the different performance between samples belonging to non-porous and porous areas was more prominent than between samples with different $v_{0}^{’}>0\%$. The mean flexural strength, ${\bar{\sigma }}_{fM}$, in the non-porous area was 544 MPa (${\bar{v}}_{f}^{’}=53.1\%$), while in the porous area it was 500 MPa (${\bar{v}}_{f}^{’}=53.6\%$), a reduction of 8.1%. Subtracting the effect of ${\bar{v}}_{f}^{’}$, the reduction in ${\bar{\sigma }}_{fM}$ increased to 9.1%. On the other hand, the mean flexural modulus, ${\bar{E}}_{f}$, was 14.7 GPa (${\bar{v}}_{f}^{’}=53.3\%$).

### 4.5. Screening of Degassing Procedures

The fractional factorial design was analyzed with respect to porosity-related attributes which showed significantly higher variability than the rest of quality-related attributes presented (Figure 12): porous area fraction, $s_{0}$, and updated void volume fraction, $v_{0}^{’}$, (only corresponding to samples CY8). In addition, time-related factors also showed significant variability. Special attention was kept on time until filling onset, $t_{0\to fill_{0}}$, time until filling end, $t_{0\to fill_{end}}$, and filling time, $t_{fill}$; because of their connection with void formation due to flow through dual-scale porous media.

In representing $s_{0}$ versus $v_{0}^{’}$, a direct linear connection between both attributes became apparent as shown in Figure 13a.

No correlation seemed to exist between $s_{0}$ and any covariate; but at confronting $v_{0}^{’}$ versus $t_{0\to fill_{0}}$, a connection between them was observed (Figure 13b). $v_{0}^{’}$ decreased as $t_{0\to fill_{0}}$ increased. Similar trends arose between $v_{0}^{’}$, and $t_{0\to fill_{end}}$ and $t_{fill}$; however, $t_{0\to fill_{0}}$ seemed to be the real cause behind these connections, since changes in resin viscosity, μ, at the onset of the filling step caused by different $t_{0\to fill_{0}}$, beyond inherent preform variability, seemed to be the real cause behind $t_{0\to fill_{end}}$ and $t_{fill}$ scatter (Figure 13c).

Factors P and T were partially correlated with $t_{0\to fill_{0}}$ too. Degassing procedures that included pressurization, high level of factor P, implied an average increment of 10 min in $t_{0\to fill_{0}}$. On the other hand, different degassing times, factor T, implied an average increment of 22 min between the high level (40 min) and the low level (20 min). Effects of $t_{0\to fill_{0}}$ on $v_{0}^{’}$ were later analyzed through ANCOVA.

The described statistical procedure was applied to the experimental data gathered in Figure 14. In Figure 15a,b, it can be seen how main factor effects $l_{N}$, $l_{S}$, and $l_{P}$ behaved similarly with both response variables; while $l_{T}$ was considerably larger in case of $v_{0}^{’}$, which could have been caused by the correlation found between $v_{0}^{’}$ and $t_{0\to fill_{0}}$, and the coupling between T and $t_{0\to fill_{0}}$. Furthermore, the NS interaction in case of $s_{0}$ (Figure 15a) stood out from the other two-factor interactions. Factors N, S, and T at high level (+) appeared to enhance void minimization, while factor P negatively affected specimen porosity.

Highlighted effects shown in Figure 16a,c were addressed through the ANOVA, performing the backward elimination until reaching the two models depicted in Table 3. Although initially unreplicated, both models evolved to replicated designs with an enough number of degrees of freedom to reliably compute error variance. It must be pointed out that, in both models, factor P was added after performing the corresponding analyses of residuals to validate both models.

In both models, there only appeared one significant term (p≤0.05), factor S in the case of $s_{0}$ and T in the case of $v_{0}^{’}$; although interaction NS and factor S were very close to the limit of significance in the case of $s_{0}$ and $v_{0}^{’}$, respectively.

The apparent relation between $v_{0}^{’}$ and $t_{0\to fill_{0}}$, the aliasing between T and $t_{0\to fill_{0}}$, the fact that T is the only main factor whose influence considerably changed between both response variables, and, finally, conducted ANOVA and ANCOVA suggested that variations in $t_{0\to fill_{0}}$ affected $v_{0}^{’}$ more than different degassing times. However, it was not possible to isolate effects of T and $t_{0\to fill_{0}}$.

Figure 16b,d shows the fitted values for the proposed models. The best performance in both cases was achieved when resin was stirred while being degassed, factor S. Besides, when none of the factors N or S were included in the degassing procedure, the results were considerably worse.

Analyses of residuals were not included to avoid a saturation of statistical graphs which do not provide any additional information from the point of view of the manufacturing process, which is the focus of this work.

## 5. Discussion

Although not being a major topic of research, previous studies have focused on the importance of resin degassing in VI manufacturing, while raising some concern about conventional vacuum degassing [16,17,37]. Air solubility can be determined at different pressure and temperature conditions; however, resin outgassing after preform filling also depends on impregnation conditions, and the interaction between resin and reinforcements [17,38,39]. Therefore, a rigorous VI processing methodology was proposed which allowed the outgassing assessment through the final porosity content of a series of specimens manufactured for that purpose.

The manufacturing procedure was based on inducing a decreasing gradient of pressure into the VI specimens, which should result in different outgassing conditions across the filling length, L. Characterizing specimens by the loss on ignition method, led to gradients of fiber volume fraction, $v_{f}^{’}$, and void volume fraction, $v_{0}^{’}$, appearing in each specimen. As reported in previous studies, trapped gradients of pressure and thickness ($\propto v_{f}^{’}$) in the laminates are closely related, requiring the former even more time to equalize during post-filling steps [24,30,31]. Furthermore, the increasing gradient of $v_{0}^{’}$ was an evidence of the presence of a continuous range of outgassing conditions into each specimen.

In all of the manufactured specimens, critical conditions at which outgassing firstly happened were enclosed into L. Setting vent pressure to $P^{vent}=-90\;\mathrm{kPa}$ along filling, post-filling and curing steps, played a key role for that purpose. A $P^{vent}$ closer to vacuum pressure could have resulted in specimens whose useful area were completely covered with porosity. The porous area fraction, $s_{0}$, can be understood as an indicator of the above mentioned critical outgassing conditions.

Although it was intended to isolate outgassing effects on void formation from flow through dual-scale porous media, variations in resin viscosity, μ, significantly affected void formation. In a future implementation of the proposed manufacturing methodology, it would be recommended to not considerably alter time until filling, $t_{0\to fill_{0}}$.

Predominant formation of inter-tow voids resulted in a fast void content increase once one enters the porous areas due to the large size of these voids. As a consequence, flexural strength, $\sigma_{fM}$, did not suffer a continuous deterioration, but a sudden drop [5,6]. In Figure 11a, two different behaviors in terms of $\sigma_{fM}$ can be identified according to the presence or not of voids in the samples. A drop of 9.1% in $\sigma_{fM}$ of porous samples (once effect of fiber content was subtracted) occurred even including the abnormally high values of some porous samples. Deterioration in $\sigma_{fM}$ was more pronounced between non-porous and porous samples than in samples with $v_{0}^{’}>0$ (Figure 11b). It is worth noting that, in the literature, the detrimental effect of porosity on other matrix-dominated mechanical properties such as inter-laminar shear strength and fatigue behavior is even more appreciable [3,4,5].

In spite of the large uncertainty in measuring $v_{0}^{’}$ through the loss on ignition method due to surface texture corrections, these measurements were more realistic than those obtained by light microscopy. The heterogeneous pore distribution did not allow to capture in a single cross-section, despite the large area analyzed, a representative picture to reliably determine the void content fraction. A more accurate quantification of void content through microscopy analysis would have required processing more cross-sections reflecting specific outgassing conditions or a volumetric measurement method [6,40,41].

After analyzing micrographic samples, inter-tow void size, $A_{0}^{inter}$, seemed to be related to the free space between tows into the preforms; hence, higher fiber content preforms should reduce $A_{0}^{inter}$ and even may decrease the total trapped void content. A similar order of pore magnitude was found in other studies focused in components manufactured by RTM [21,42]. In order to be really aware of the problem, it is highlighted that the maximum values of $A_{0}^{inter}$ and Feret’s diameter, $\emptyset_{Feret}$, found in the micrographic samples, $0.41\;{\mathrm{mm}}^{2}$ and 3.09 mm, respectively (Figure 10a). The maximum $\emptyset_{Feret}$ was even larger than the thickness of the specimens.

The screening experiment confirmed the concern about traditional vacuum degassing. It has been proved that mechanisms to enhance bubble formation are fundamental to perform effective resin degassing. Assisting conventional vacuum degassing by adding of nucleation media and/or HS resin stirring has arisen as a real alternative to minimize outgassing in VI and enhance dissolution of voids formed during preform filling. Furthermore, both involved degassing times, $t_{deg}$, were long enough to not affect resin outgassing; whereas, later resin pressurization, to remove micro-bubbles trapped near resin surface, counter-productively resulted in higher void contents.

The apparent degassing performance was similar in all degassing procedures. Initially, bubble clusters were formed at the free surface and the average bubble size increased due to bubble coalescence and diffusion of air molecules, resulting in the increment of the volume of resin. Then, the volume of resin reached a maximum level, but bubble continued increasing in size. After a short period of time, the volume diminished to its initial level as the average bubble size also decreased. Finally, non-clustered bubbles burst at the free surface of the resin, while the volume kept close to the initial level. The described process did not require more than 20 min in any case, and the quantity of bubbles trapped near the free surface did not significantly vary between experiments; therefore, a false impression could have been created if attention had been only paid to resin behavior during degassing.

The best results in terms of porous area minimization were obtained when only HS resin stirring was involved in the degassing procedure. Apart from being an easily implementable degassing procedure, it did not involve waste of any additional material as in the case of the nucleation medium. The combination of HS resin stirring and nucleation medium showed a worse result than when only stirring was involved in the degassing procedure. It may be explained by a higher rotation resistance at placing the magnetic rod over the nucleation medium, involving a reduction in the stirring speed.

Future work on this research should include the analysis of the effect of stirring speed and the influence of more stirring points on degassing efficacy. Furthermore, a pressure measurement system would be useful to monitor inlet pressure evolution after clamping the inlet; since preform thickness measurement during post-filling did not provide absolute data about the gradient of pressure, it allowed, nevertheless, a qualitative comparison between different specimens.

## 6. Conclusions

The effects of some factors (addition of a nucleation medium, HS resin stirring, and later pressurization and degassing time) on conventional vacuum degassing have been analyzed through a screening experiment based on a fractional factorial design, with the aim of finding a really effective degassing procedure to minimize porosity in specimens manufactured by VI. A detailed VI process has been defined to allow the direct assessment of degassing efficacy by means of the resin outgassing behavior though final specimen void content.

Specimen characterization revealed a large magnitude of inter-tow voids, finding pores even larger than specimen thickness which resulted in a sudden and significant drop in flexural strength. The results of the screening experiment supported the idea that conventional vacuum degassing is not really effective and mechanisms to enhance bubble formation are fundamental. Stirring the resin while being degassed at ≈300 rpm arose as an easily implementable and significantly efficient procedure of reducing final specimen void content.

These findings are not only crucial to approach the final goal of manufacturing porosity-free VI parts; but conducting a proper degassing procedure is fundamental in the analysis of matrix-dominated mechanical properties in VI samples, since the presence of undesired porosity avoids taking advantage of the full potential of FRPs.

Finally, it is worth pointing out that the results obtained and the conclusions drawn from the present study are only qualitatively applicable to other experimental conditions, because outgassing behaviour is closely related to the materials involved and the VI governing parameters. Moreover, further research is necessary to evaluate the effect of higher stirring speeds and more stirring points on degassing efficacy.

## Figures and Tables

**Figure 1 polymers-13-02876-f001:**
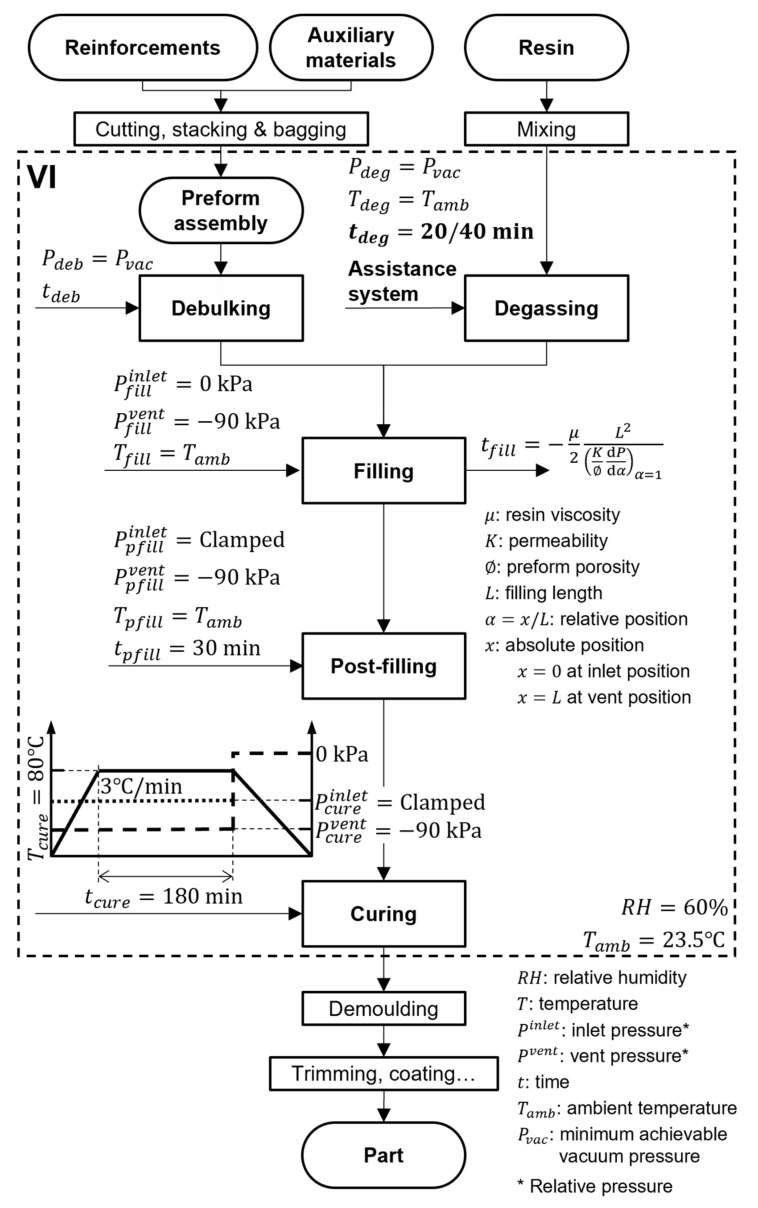
Flow chart and governing parameters of the VI process.

**Figure 2 polymers-13-02876-f002:**
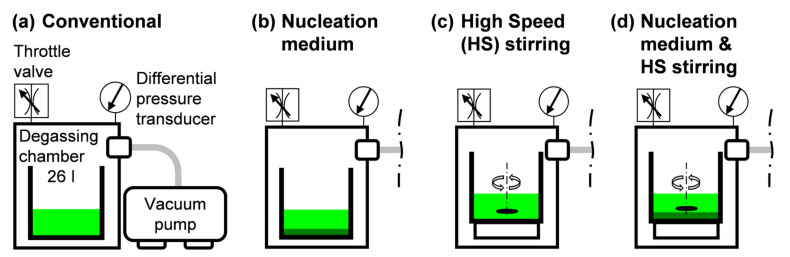
Vacuum degassing configurations: (**a**) Conventional vacuum degassing without additional assistance; (**b**) Assisted by adding a nucleation medium; (**c**) Assisted by high speed resin stirring; (**d**) Assisted by adding a nucleation medium and high speed resin stirring.

**Figure 3 polymers-13-02876-f003:**
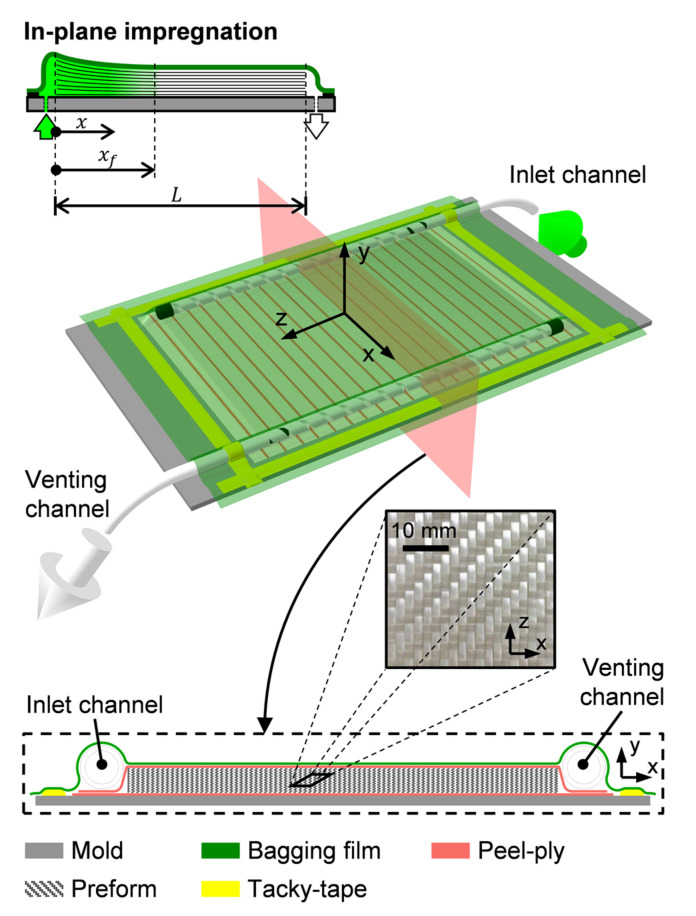
VI preform assembly (schematic dimensions are representative).

**Figure 4 polymers-13-02876-f004:**
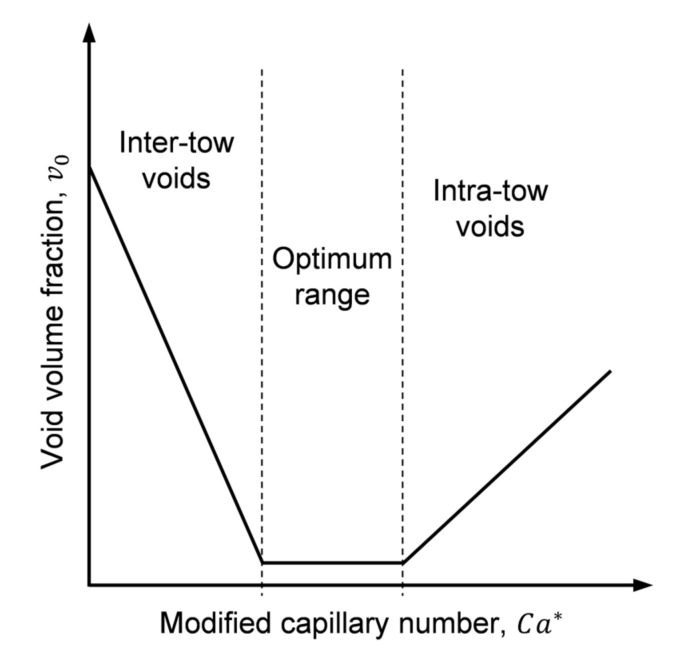
Void formation due to resin flow through heterogeneous dual-scale porous media.

**Figure 5 polymers-13-02876-f005:**
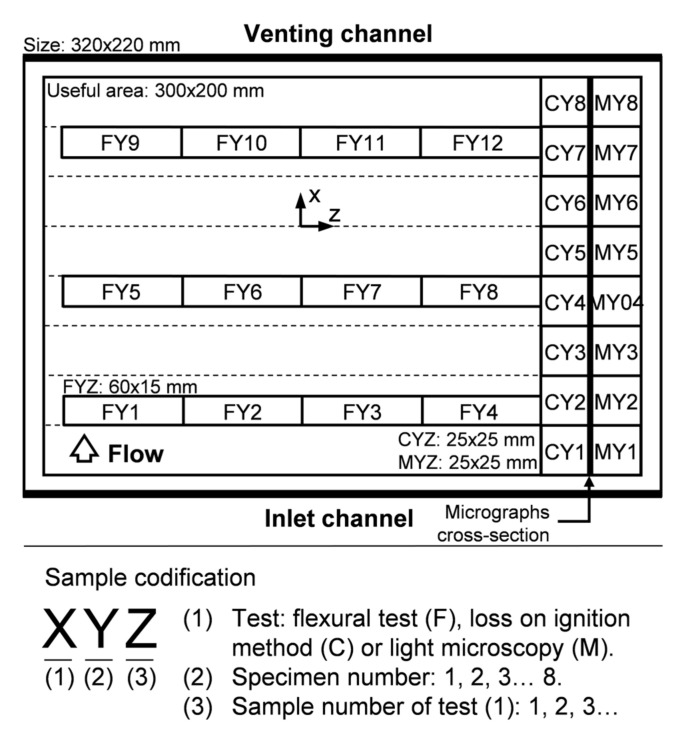
Generic pattern of samples: size, distribution, and codification rules.

**Figure 6 polymers-13-02876-f006:**
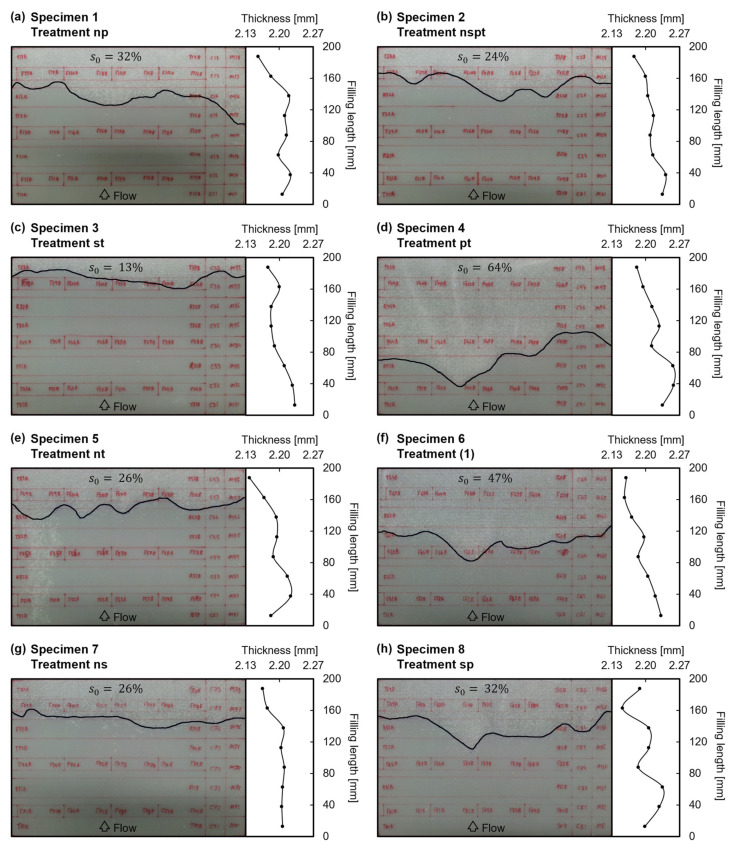
Scaled photographs of the manufactured specimens’ useful area (300 mm × 200 mm): (**a**) Specimen 1, treatment np; (**b**) Specimen 2, treatment nspt; (**c**) Specimen 3, treatment st; (**d**) Specimen 4, treatment pt; (**e**) Specimen 5, treatment nt; (**f**) Specimen 6, treatment (1); (**g**) Specimen 7, treatment ns; (**h**) Specimen 8, treatment sp. Samples marked in red on specimens’ top face; boundary of the porous area fraction, $s_{0}$, digitally underlined in black; and thickness profile measured at samples MYZ along the filling length.

**Figure 7 polymers-13-02876-f007:**
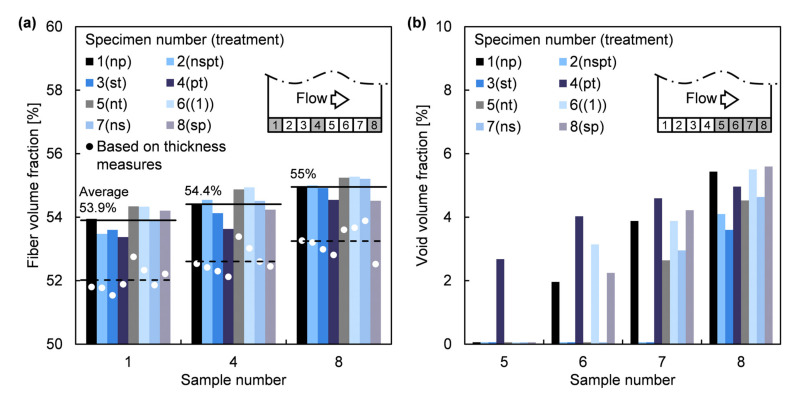
Results of the loss on ignition method: (**a**) Fiber volume fraction, $v_{f}^{’}$; and (**b**) Void volume fraction, $v_{0}^{’}$.

**Figure 8 polymers-13-02876-f008:**
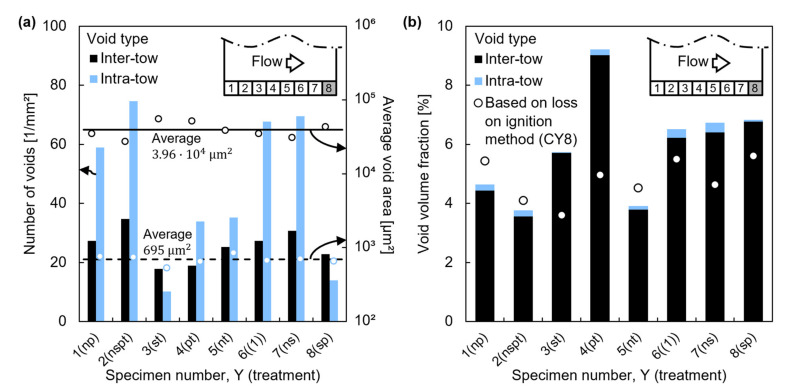
Size, occurrence, and total inter- and intra-tow void content: (**a**) Voids occurrence and average void area, $A_{0}$; (**b**) Void area fraction, $a_{0}$.

**Figure 9 polymers-13-02876-f009:**
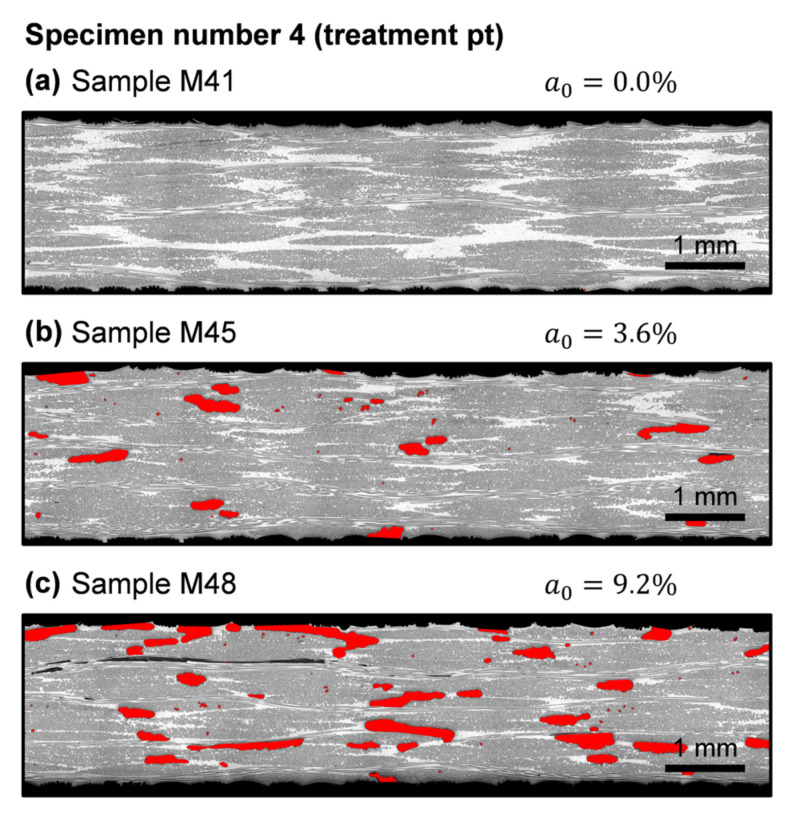
Detail views of micrographic samples belonging to specimen number 4 and void area fraction, $a_{0}$, of the corresponding whole sample (not only the partial cross-section depicted): (**a**) Sample M41; (**b**) Sample M45; (**c**) Sample M48.

**Figure 10 polymers-13-02876-f010:**
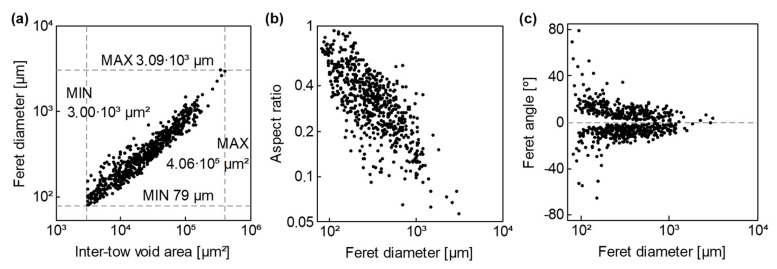
Shape descriptors of inter-tow voids: (**a**) Feret’s diameter, $\emptyset_{Feret}$; (**b**) Aspect ratio, AR; (**c**) Feret’s angle, $\varphi_{Feret}$.

**Figure 11 polymers-13-02876-f011:**
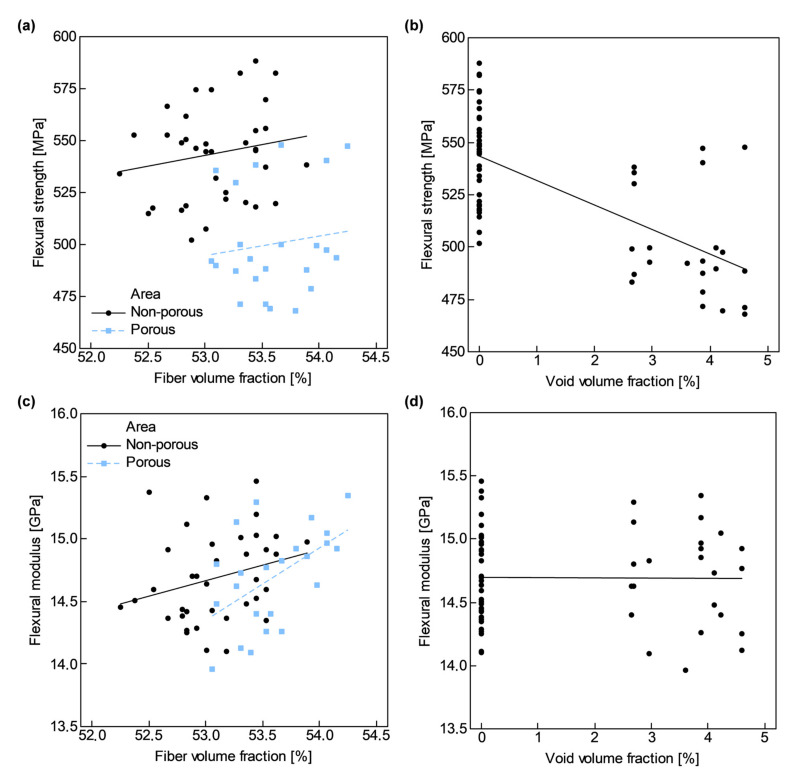
Flexural properties versus estimated fiber volume fraction, ${\hat{v}}_{f}^{’}$, and void volume fraction, $v_{0}^{’}$: (**a**,**b**) Flexural strength, $\sigma_{fM}$; and (**c**,**d**) Flexural modulus, $E_{f}$.

**Figure 12 polymers-13-02876-f012:**
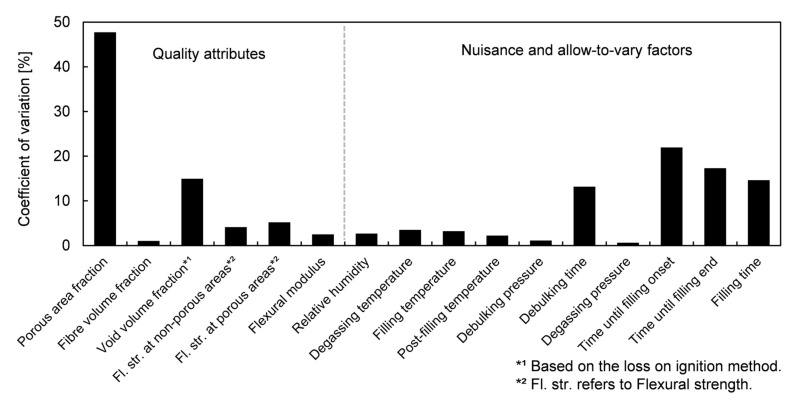
Coefficients of variation of measured quality attributes and monitored nuisance and allow-to-vary factors.

**Figure 13 polymers-13-02876-f013:**
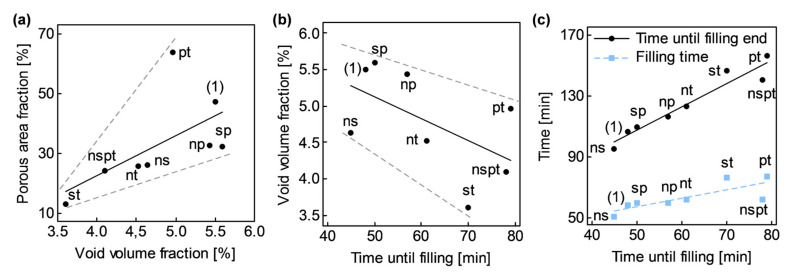
Noteworthy dependencies between response variables and covariates: (**a**) Porous area fraction, $s_{0}$, vs. void volume fraction, $v_{0}^{’}$; (**b**) Time until filling, $t_{0\to fill_{0}}$; (**c**) Time until filling end, $t_{0\to fill_{end}}$, and filling time, $t_{fill}$.

**Figure 14 polymers-13-02876-f014:**
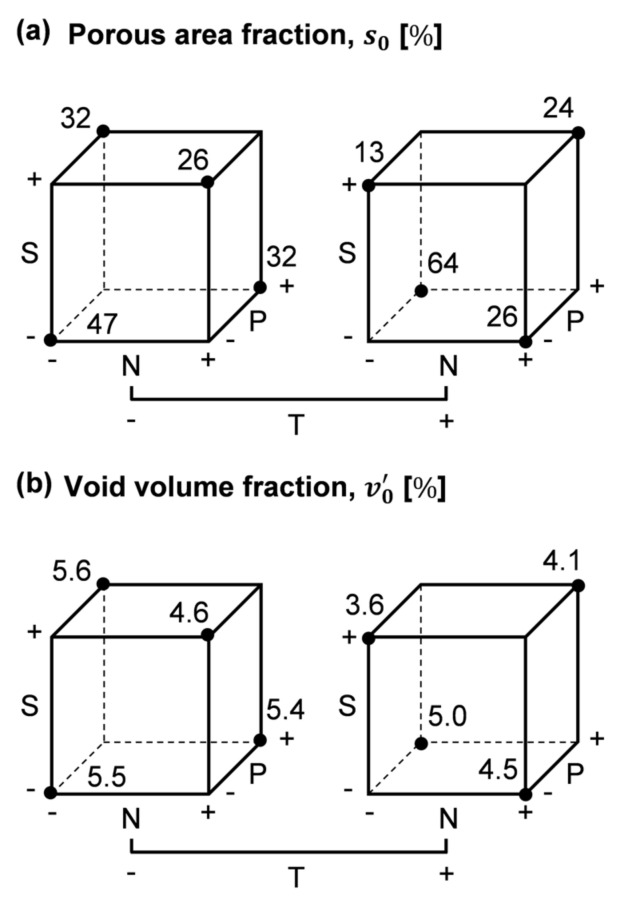
Cube plots of the input data of the screening experiment: (**a**) Porous area fraction, $s_{0}$; (**b**) Void volume fraction, $v_{0}^{’}$.

**Figure 15 polymers-13-02876-f015:**
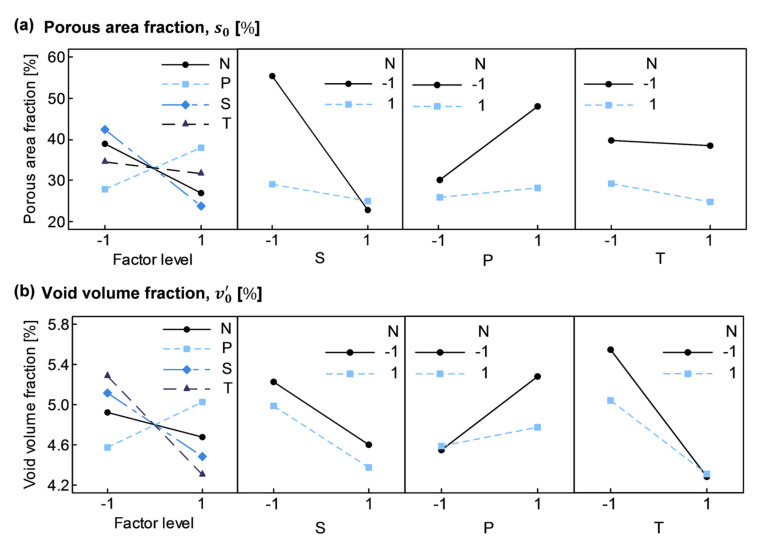
Effects of main factors and two-factor interactions (interaction effects are aliased according to $l_{NS}=l_{PT}$, $l_{NP}=l_{ST}$, and $l_{NT}=l_{SP}$): (**a**) Porous area fraction, $s_{0}$; (**b**) Void volume fraction, $v_{0}^{’}$.

**Figure 16 polymers-13-02876-f016:**
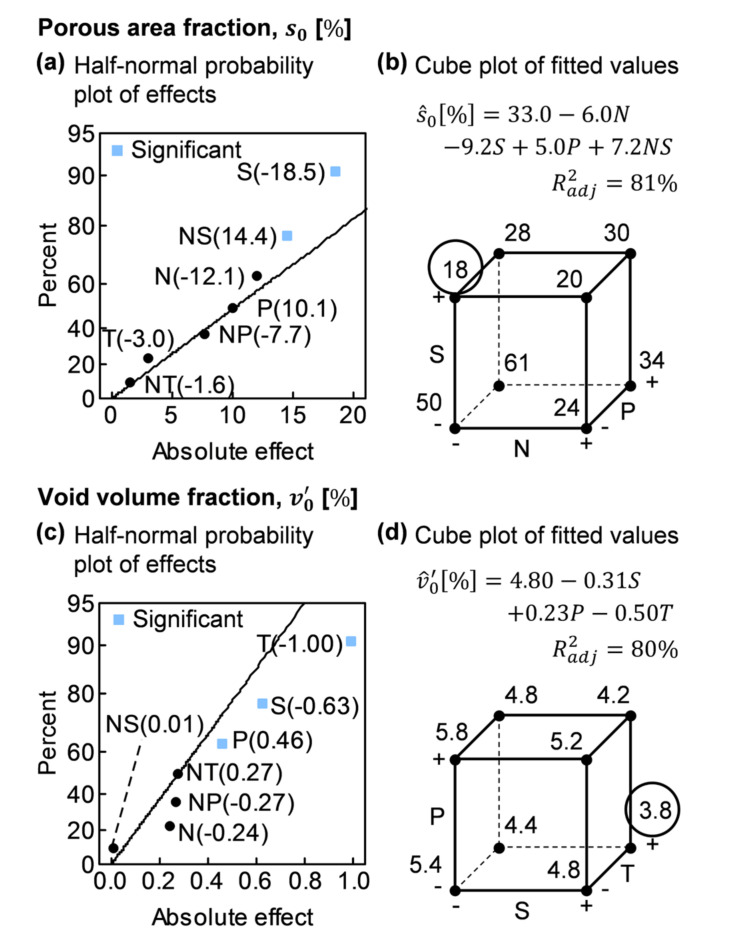
ANOVA supporting plots: (**a**) Porous area fraction, $s_{0}$, half-normal probability plot of effects; (**b**) $s_{0}$ cube plot of fitted values; (**c**) Void volume fraction, $v_{0}^{’}$, half-normal probability plot of effects; (**d**) $v_{0}^{’}$ cube plot of fitted values.

**Table 1 polymers-13-02876-t001:** Design factors of the screening experiment of degassing procedures.

Factor	Level ^1^		Details	Motivation
	High (+)	Low (−)		
Nucleation medium (N)	On	Off	Scotch-Brite	Enhancing heterogeneous bubble formation
HS stirring (S)	On	Off	Magnetic stirrer at 300 rpm with a rod of 40 mm	Enhancing bubble formation due to cavitation
Pressurization (P)	On	Off	200 kPa ^2^ for 5 min	Collapse of micro-bubbles [16]
Degassing time (T)	40 min	20 min	-	-

^1^ In regression models, high and low levels were considered +1 and −1, respectively. ^2^ Relative pressure.

**Table 2 polymers-13-02876-t002:** The $2_{\mathrm{IV}}^{4-1}$ fractional factorial design, principal one-half fraction, applied to the screening experiment of degassing procedures.

Specimen Number (Y)	Basic Design			T = NSP	Treatment
	N	S	P		
6	−	−	−	−	(1)
5	+	−	−	+	nt
3	−	+	−	+	st
7	+	+	−	−	ns
4	−	−	+	+	pt
1	+	−	+	−	np
8	−	+	+	−	sp
2	+	+	+	+	nspt

**Table 3 polymers-13-02876-t003:** ANOVA tables including Sum of Squares (SS); Degrees of Freedom (DF); Mean Squares (MS); F-statistic, $F_{0}$, associated to each term; and level of significance, p, associated to each F-statistic.

Source	SS	DF	MS	$F_{0}$	*p*
Porous area fraction, $s_{0}\left[\%\right]$			$R_{adj}^{2}=81\%$ ^1^		
N	291.6	1	291.6	6.19	0.089
S	683.1	1	683.1	14,5	0.032
P	203.3	1	203.3	4.32	0.129
NS	416.4	1	416.4	8.84	0.59
Error	141.3	3	47.1		
Total	1736	7			
Void volume fraction, $v_{0}^{’}\left[\%\right]$			$R_{adj}^{2}=80\%$ ^1^		
S	0.78	1	0.78	7.60	0.051
P	0.42	1	0.42	4.04	0.115
T	1.98	1	1.98	19.2	0.012
Error	0.41	4	0.10		
Total	3.59	7			

^1^ Adjusted coefficient of determination, $R_{adj}^{2}$, is appropriate when available DF to compute error variance is small.

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
