# Peer review of "Void Content Minimization in Vacuum Infusion (VI) via Effective Degassing"

_polymers, 2021, doi:10.3390/polym13172876_

Round 1

Reviewer 1 Report

I found this paper very technical and difficult to read. The paper does not give more insight on the reasons of the void formation in the resins degassed with different methods. By contrast, it is overcharged with technical considerations about the degassing mold filling process. I believe that the research itself is well-done and precisely described, and though this paper may be suitable for a more specialized journal.

Author Response

First of all, thanks for reviewing our work. It is totally true that the introduction sections were focused on the particularities that influence void formation during vacuum infusion, and how these particularities drove the experiment design to allow a fair assessment between degassing techniques. Since the concepts behind heterogeneous nucleation and cavitation are well known across a variety of fields, we decided to keep that part of the introduction brief to prevent the manuscript exceeding a reasonable length.

Reviewer 2 Report

The article presents a detailed investigation of degassing effects on void content minimization in vacuum infusion process. The article is very well structured and technically sound. Below are a few comments and suggestions:

  1. In the discussion section (section 5), the authors recommended that considerably not altering time until filling would lead to lower variations in resin viscosity and hence void formation (lines 505-506, page 20). This statement contradicts findings from the authors since the st configuration had the best results with lowest porosity as noted by them in lines 552-555, page 21, 577-579, page 22. It would be great if the authors could provide a justification for this recommendation.
  2. Lines 433-434 on page 16 can be rephrased for better understanding
  3. Figure 6 page 11, it is recommended that authors provide axis title for filling length as well (atleast on the right hand side of 16-b,d,f,h)
  4. Line 223 Page 7, it would be easier if the authors provide nomenclature for void volume fraction near the symbol v0
  5. line 93, page 2, figure call out is missing

Author Response

Thanks for the thorough review of our work. Below, all five suggestions are individually addressed:

  1. The reasoning behind that recommendation was explained in detail in lines 471-475. However, using the term ‘porosity’ in line 555 might be misleading, so it has been replaced by ‘porous area’.
  2. Those lines have been rephrased in an attempt to provide better clarity.
  3. Filling length axis titles have been added as suggested.
  4. Nomenclature has been added for both vf and v0.
  5. Figure callout has been added.

Reviewer 3 Report

This paper deals with the porosity characterization in the vacuum infusion process of polymer composite production. The experiments are detailed and a great deal of data were presented. Some fundamental knowledge is also introduced. It is acceptable for publication in this journal.

Fig. 6 contains some color background. Just make sure this is an intentionally built feature.

Author Response

Thanks for your appreciations.

Figure 6 should only have a white background behind the manufactured specimens and the thickness plots. In the uploaded word file I cannot see anything awkward, but I have had to upload this picture again because of another reviewer’s suggestion. Please, let me know if you still see that color background.